# Effects of Methotrexate and Tofacitinib on Mitochondrial Function and Oxidative Stress in Human Synovial Cells In Vitro

**DOI:** 10.3390/ijms26178173

**Published:** 2025-08-22

**Authors:** Valentina Mihaylova, Desislav Tomov, Rositsa Karalilova, Zguro Batalov, Anastas Batalov, Victoria Sarafian, Maria Kazakova

**Affiliations:** 1Department of Medical Biology, Medical University, Blvd. Vasil Aprilov 15A, 4000 Plovdiv, Bulgaria; mihailova83@abv.bg (V.M.);; 2Research Institute, Medical University, 4002 Plovdiv, Bulgaria; 3Department of Propaedeutics of Internal Diseases, Medical University, 4002 Plovdiv, Bulgaria; 4Clinic of Rheumatology, University Hospital “Kaspela”, 4001 Plovdiv, Bulgaria

**Keywords:** SW982, mitochondrial function, oxidative stress, ISOPGF2A

## Abstract

Rheumatoid arthritis (RA) is an autoimmune disease affecting the synovium. Mitochondrial dysfunction is considered a critical factor in the pathogenesis of RA. The aim of the study was to determine the effect of methotrexate and tofacitinib on mitochondrial function and oxidative stress in an in vitro study on the model synovial cell line SW982. TNF-alpha-stimulated SW982 cells, as well as control untreated cells, were incubated with methotrexate and tofacitinib. A metabolic test was performed to assess mitochondrial function. The oxidative stress generated after the application of the therapeutics was determined by a chromatographic analysis. The results obtained showed an increase in ATP levels (*p* < 0.0001) and a decrease in proton leak (*p* < 0.0003) after treatment with tofacitinib. The opposite trend was observed—reduced ATP production (*p* < 0.0096) and increased levels of proton leak (*p* < 0.0001)—after treatment with methotrexate. A two-fold increase in 8-ISOPGF2A was measured in comparison to TNF-alpha-stimulated and untreated cells. The dynamics of mitochondrial activity and oxidative stress were monitored in a certified RA model cell line after the administration of two different therapeutics. Methotrexate was found to induce mitochondrial dysfunction and oxidative stress in vitro, while tofacitinib partially improved mitochondrial parameters.

## 1. Introduction

Rheumatoid arthritis (RA) is an inflammatory autoimmune disease affecting the synovium. It is characterized by pannus formation and cartilage and joint destruction [1]. Despite the significant progress in the treatment of RA in recent years, its etiology is still unclear, and the mechanisms involved in the pathogenesis have not been studied in detail yet. To face these challenges intensive scientific work is required to improve patients’ outcomes and quality of life.

There are hypotheses about the occurrence of unregulated citrullination, which provokes the formation of anti-citrullinated protein antibodies (ACPA) [2]. In addition, disorders in the joints can lead to the production of proinflammatory cytokines, which trigger destructive changes and modification of self-antigens [3]. The heritability of RA is determined to be about 65% for seropositive patients and around 20% for seronegative carriers of RA. Advances in genetic technologies and the appropriate stratification of cohorts have boosted the understanding of disease genetics. Over a hundred loci with mainly immune functions have been shown to be related to the development of RA. It is believed that the HLA-DRB1 system is mainly involved in the onset and development of the disease [4].

A critical factor in the pathogenesis of RA is the presence of mitochondrial dysfunction. Altered mitochondrial activity in synovial cells and chondrocytes inevitably leads to disease promotion [5]. In the presence of mitochondrial dysfunction, the balance between glycolysis and oxidative phosphorylation is disturbed, which affects the amount of adenosine triphosphate (ATP) produced. Thus, in turn irreversible metabolic disorders follow [6]. The deficiency of ATP hampers the regeneration of cartilage tissue. Mitochondrial damage is also associated with high proton leakage, respectively, and increased levels of mitochondrial reactive oxygen species (mtROS) [7]. They seriously damage cell structures and lead to a high probability of mitochondrial DNA mutations. Damaged mitochondria can trigger a strong immune response. Increased superoxide flux resulting from these mitochondrial DNA lesions may subsequently contribute to metabolic oxidative stress, genomic instability, and cellular damage [8]. Reactive oxygen species (ROS) can oxidize lipids, which subsequently undergo fragmentation to produce 8-isoPGF2alpha, which is one of the most extensively studied isoprostanes and the most commonly used biomarker to assess oxidative stress [9].

Current therapies for RA treatment include nonsteroidal anti-inflammatory drugs (NSAIDs) and disease-modifying anti-rheumatic drugs (DMARDs) as first-line therapies for all newly diagnosed cases of RA, and biological response modifiers, which are targeted agents that selectively inhibit specific immune system molecules [10]. Methotrexate (MTX) and tofacitinib (TFB) have a central role in RA therapy, representing distinct but clinically complementary treatment strategies. While MTX is the gold standard csDMARD with broad anti-inflammatory effects [11], TFB offers a more targeted approach by directly interacting with cytokine-mediated intracellular signaling [12]. A comparative analysis of their effects in a synovial cell model may provide mechanistic insight into the differential regulation of inflammatory pathways, with potential relevance for optimizing treatment selection, sequencing, or combination strategies in RA. Furthermore, both drugs are included in major RA treatment guidelines (e.g., EULAR, ACR), additionally supporting their translational effectiveness.

We present novel data on mitochondrial function and oxidative stress in RA based on a study-certified synoviocytes cell line after treatment with two therapeutics used in clinical settings—MTX and TFB. Dynamics in proton leak, ATP production, and 8-isoprostglandin F2 alpha (8-isoPGF2alpha) levels are reported. These findings may provide useful tools for monitoring therapeutic effects in RA patients.

## 2. Results

Following the aim of the study to determine mitochondrial function and oxidative stress, we started with a MTT assay to determine treatment concentrations on the synovial cell line SW982.

### 2.1. Cell Viability (MTT—Test)

SW982 cells were treated with different concentrations of MTX and TFB for 24 h after stimulation with 10 ng/mL TNF-alpha for 24 h. Cell viability after both treatments (Figure 1a,b) varied between 86 and 100%, which proved the lack of cytotoxicity of the therapeutics at the concentrations used on the cell model. In addition to the MTT assay, we also performed a cell viability analysis using an automated cell analyzer, LUNA. The results showed a comparable dose-dependent trend between MTT and trypan blue measurements, reinforcing the validity of our observations.

### 2.2. Mitochondrial Function (Mito Stress Test)

In order to determine how the applied therapeutics affect the mitochondrial function in the synovial cell line, the Mito Stress Test was carried out. The data showed a reduced ATP production (Coupling efficiency (%)) (mean difference: 7.2, 95% CI: 2.31–12.09, and *p* < 0.0096) (Figure 2a) and increased levels of proton leak (mean difference: 8.04, 95% CI: 5.90–10.90, and *p* < 0.0001) (Figure 2b) in MTX-treated cells compared to cells stimulated with TNF-alpha only.

Legend (a): Results are presented as mean ± SD; Coupling efficiency % is the ATP production in %, normalized on the basal respiration. Blue—control SW982 cells (untreated, unstimulated), red—TNF-alpha-stimulated SW982 cells and TNF-alpha-stimulated SW982 cells, treated with MTX—1 µg/mL.

Legend (b): Results are presented as mean ± SD; blue—control SW982 cells (untreated, unstimulated); and red—TNF-alpha-stimulated SW982 cells and TNF-alpha-stimulated SW982 cells, treated with MTX—1 µg/mL.

The opposite trend of an increased level of ATP (Coupling efficiency (%)) (mean difference: 11.2 95% CI: 7.39–15.01, and *p* < 0.0001) (Figure 3a) and decrease in proton leak (mean difference: 3.07, 95% CI: 2.62–3.52, and *p* < 0.0003) (Figure 3b) was observed in cells treated with TFB.

Legend (a): Results are presented as mean ± SD; Coupling efficiency % is the ATP production in %, normalized to the basal respiration; blue—control SW982 cells (untreated, unstimulated); and red—TNF-alpha-stimulated SW982 cells and TNF-alpha-stimulated SW982 cells, treated with TFB—100 nM.

Legend (b): Results are presented as mean ± SD; blue—control SW982 cells (untreated, unstimulated); and red—TNF-alpha-stimulated SW982 cells and TNF-alpha-stimulated SW982 cells, treated with TFB—100 nM.

Representative curves of mitochondrial respiration in SW982 cells are presented in Figure 4a,b.

Legend (a): Blue line—control SW982 cells (untreated, unstimulated), red line—TNF-alpha-stimulated SW982 cells, and green line—TNF-alpha-stimulated SW982 cells, treated with MTX.

Legend (b): Blue line—control SW982 cells (untreated, unstimulated), red line—TNF-alpha-stimulated SW982 cells, and green line—TNF-alpha-stimulated SW982 cells, treated with TFB. The application of the three inhibitors allowed for the calculation of the ATP, maximal and reserve respiratory capacity, proton efflux, and non-mitochondrial respiration.

### 2.3. Oxidative Stress (Chromatographic Analysis for 8-IsoPGF2alpha Detection)

Looking for a relationship between the impaired mitochondrial dynamics and the changes in proton leak, the level of oxidative stress was assessed by examining the marker 8-isoPGF2alpha.

An increase of 8-isoPGF2alpha was found only in supernatants from TNF-alpha-stimulated and MTX-treated cells (mean difference: 266.4, 95% CI: 250.9–281.9, and *p* < 0.0001) (Figure 5). No statistically significant difference was observed in the other groups. The chromatograms can be seen in the Appendix A.

The graphics on Figure 5 represent the chromatographic results for 8-isoPGF2alpha.

Legend: Results are presented as mean ± SD; blue—control SW982 cells (untreated, unstimulated); red—TNF-alpha-stimulated SW982 cells; green—TNF-alpha-stimulated SW982 cells, treated with 1 µg/mL MTX; and purple—TNF-alpha-stimulated SW982 cells, treated with 100 nM TFB.

## 3. Discussion

The disturbed immunometabolism and involvement of mitochondria in the pathogenesis of RA have been proven in recent studies [13]. It has been proven that any damage of mitochondrial function can trigger a series of cascading events that lead to the development of the disease [5]. Research in this direction may shed additional light on the understanding and control of the disease. Management of excessive mitochondrial oxidative stress may offer new targets for both the prevention and therapy of RA [6].

To examine how first-line therapy drugs affect mitochondrial status and oxidative stress, we established an experimental design model using the synovial cell line SW982. TNF-alpha stimulation aimed to recreate an inflammatory model similar to that seen in RA in vivo [14]. MTX, a folic acid antagonist, is a widely used medication to treat RA patients [15]. In the present study, this therapeutic medication was appliedto treat a cell line. When measuring its effect on mitochondrial parameters, we found an increased proton leak. This condition leads to the activation of glycolysis, disturbed structure of the inner mitochondrial membrane, as well as to increased oxidative stress [16]. After treatment of TNF-alpha-stimulated synovial cells with MTX, ATP production was statistically lower compared to the TNF-alpha-stimulated cell line. The result is likely to be due to the property of the therapeutic medication to inhibit complex III of the electron transport chain [17]. Analogously, the same parameters were examined when the TNF-alpha-stimulated cells were treated with TFB. This drug inhibits the JAK-STAT pathway, reducing the inflammatory response [18]. The results obtained in these experimental settings showed the opposite trend. There was a higher ATP production and a drop in proton leak. This is not surprising, since JAK-STAT inhibitors catalyze oxidative phosphorylation, which leads to the reduction in free radicals, shift from glycolytic to oxidative metabolism, and repair of the inner mitochondrial membrane [19]. The same trend was found in our previous studies, where the experiment was conducted in PBMCs isolated from RA patients treated with MTX and TFB [20].

To determine the direct effect of MTX and TFB on oxidative stress, we assessed 8-isoPGF2alpha in the supernatant of treated synoviocytes. The marker has been applied to measure oxidative stress in over 500 animal studies and 900 human studies [21]. Elevated 8-isoPGF2alpha is found in various pathologies such as generalized anxiety disorder, preeclampsia, diabetes, and others compared to healthy controls [22,23,24].

In our study 8-isoPGF2alpha showed a two-fold increase after the administration of MTX. Schmidt, S. et al. reported that MTX induced oxidative stress by depleting glutathione. The same authors also demonstrated impaired mitochondrial respiration in the HepaRG cell line treated with MTX [25]. Another group showed that five days after the treatment of rats with MTX, the levels of serum markers of oxidative stress were significantly increased [26].

Despite the established dynamics in the effect of MTX and TFB, our study has its limitations. They are related to the cell model, on which the therapeutics are tested, as well as to the time periods of the treatments. The use of a single time point (24 h) for TNF-alpha exposure models acute inflammation but does not capture the temporal dynamics of the cellular response. Future studies incorporating a full time-course (e.g., 6, 48, and 72 h) will be important to better understand both early and late phases of inflammatory response. It would be informative to follow the action of other pharmacological groups of therapeutics used in RA on the same experimental model.

## 4. Materials and Methods

The activity of MTX and TFB was assessed in vitro in the synovial sarcoma cell line SW982.

### 4.1. Cell Culture

Human synovial sarcoma cells SW-982, ATCC (LOT:70054119), were cultured in Leibovitz’s L-15 medium, Sigma Aldrich, St. Louis, MO, USA, (Cat No. 30-2008, LOT:0000282233) with 10% FBS, Sigma Aldrich (LOT:0001660391) and 1% Penicillin/Streptomycin, Biotech (LOT:763018). The L-15 medium is designed for CO_2_-independent cultures, and therefore, cells were maintained at 37 °C in a humidified incubator without CO_2_. The medium was replaced every 2–3 days. Cells from passages 3 to 7 were used for the experiments. SW982 is a human synovial cell line derived from the synovial tissue of the knee joint. It exhibits several phenotypic and functional similarities to fibroblast-like synoviocytes (FLS) isolated from patients with RA. SW982 cells express key cytokines and mediators associated with RA pathology, such as IL-6, IL-8, TNF-alpha, and matrix metalloproteinases (MMPs) like MMP-1 and MMP-3, which mediate cartilage destruction in RA. SW982 responds robustly to TNF-α, IL-1β, and LPS, which mimic the inflammatory environment of the RA joint. Upon stimulation, it activates NF-κB and MAPK pathways—major drivers in RA synoviocytes’ signaling [27]. Unlike primary RA FLS, SW982 cells are immortalized and provide a reproducible, consistent, and cost-effective model for high-throughput drug screening and mechanistic studies.

### 4.2. Cell Viability Assay

In order to select the exact concentrations for the subsequent assessment of the effect of MTX and TFB on mitochondrial function and oxidative stress in the cell model, we performed the MTT assay. It is a colorimetric test used to measure cell viability, proliferation, and cytotoxicity. Metabolically active living cells transform the yellow tetrazolium salt MTT into purple formazan crystals soluble in DMSO catalyzed by mitochondrial dehydrogenases. SW982 cells were seeded at a density of 2 × 10^4^ cells/well in 96-well culture plates. TNF-alpha was used to induce inflammation in the cell line to mimic an RA model.

Cells were cultured in triplicate for 24 h and then incubated with TNF-alpha at a concentration of 10 ng/mL for the next 24 h. This was followed by the administration of MTX and TFB at six different concentrations for 24 h. All data were normalized to the L-15 medium used as a solvent. The design of the experiment is shown in Figure 6.

After treatment with the drugs, the MTT solution was added to the wells at a final concentration of 0.5 mg/mL, and cells were incubated for 3 h at 100% atmospheric air at 37 °C. Absorbance was measured at 490 nm in a Tecan sunrise plate reader. The absorbance of the control untreated cells was 100%, while the cell viability of the treated samples was counted using the formula: % = (A from the experimental wells/A from the control wells) × 100%. After preliminary studies with different concentrations of MTX and TFB, the concentrations we used for the next experiments were 1 µg/mL MTX and 100 nM TFB. Both concentrations were consistent with the permitted in vivo administrations of the therapeutics [28] and ensured cell vitality of about 87%.

In addition to the MTT assay, we also performed a cell viability analysis using an automated cell analyzer, LUNA, based on the trypan blue exclusion, which measures membrane integrity as a complementary and independent viability marker.

### 4.3. Treatments with MTX and TFB

The concentrations of both drugs were aligned with the published data [28]. They were pursuant at the license dose to mimic the in vivo administration. We extended their range by adding two more concentrations to the lowest and to the highest concentrations cited in the available literature.

The concentration of 1 µg/mL MTX was selected based on published pharmacokinetic data in patients with RA. After low-dose administration (e.g., 15 mg/week orally or subcutaneously), reported plasma Cmax values ranged from 0.5 to 2 µg/mL. Importantly, MTX concentrations in synovial fluid closely mirror those in plasma, and values around 1 µg/mL have been detected within a few hours of dosing, with prolonged retention in the joint space due to slower clearance [29]. Therefore, 1 µg/mL represents a clinically relevant synovial concentration and is appropriate for modeling local drug exposure in inflamed joints.

MTX was purchased from Cayman (Cat № 22378) and was used in the following concentrations: 10; 1; 0.1; 0.01; 0.001; and 0.0001 µg/mL.

The concentration of 100 nM TFB was chosen based on its pharmacokinetic profile and reported in vitro activity. Following therapeutic dosing (e.g., 5–10 mg twice daily), plasma Cmax values typically range from 100 to 160 ng/mL, corresponding to approximately 240–380 nM. Moreover, in vitro studies have shown effective inhibition of JAK-STAT signaling at 50–200 nM, depending on the cell type [28]. Thus, 100 nM TFB falls within the therapeutic range and is suitable for simulating achievable physiological exposure.

TFB was provided from Cayman (Cat № 25046) and was applied in ioncentrations: 200; 100; 50; 25; 12.5; and 6.13 nM.

### 4.4. Detection of Mitochondrial Function

The metabolic test detects oxygen consumption (OCR), an indicator of the mitochondrial respiration of living cells in real time. The bioenergetic profile is determined by five parameters such as ATP production, proton efflux, maximal respiratory capacity (MRC), reserve respiratory capacity (SRC), and non-mitochondrial oxygen consumption (NMOC) on the Seahorse analyzer XFe96 (Agilent Technologies, Santa Clara, CA, USA). After the assessment of cell viability and cell numbers, plates were seeded at a concentration of 2 × 10^4^ cells/mL in Leibovitz L-15 basal medium. The next day, the cell culture was visualized using an inverted microscope to verify cells’ stable distribution and adherence to the wells. A part of the cells was then treated for 24 h with 10 ng/mL TNF-alpha to induce inflammation. The therapeutics were added after 24 h. The cultivation with the drugs continued for another 24 h. Immediately prior to analysis, the basal culture medium was replaced with the RPMI1640–Seahorse. The test was conducted three times in a consecutive series with the strictest implementation of the same experimental procedure. Mitochondrial respiration was examined in real time after application of the inhibitors. Basal OCR measurements were performed before ATP synthase was inhibited using oligomycin 1.0 µM to detect mitochondrial ATP production. After the treatment of cells with FCCP 1.5 µM, maximal mitochondrial respiration and reserve respiratory capacity were determined. Finally, the addition of rotenone and antimycin A 0.5 µM allowed for the assessment of non-mitochondrial respiration rates. The data from all experiments were statistically processed.

### 4.5. High-Performance Liquid Chromatography with Mass Detection for 8-IsoPGF2alpha

Cells were seeded in 6-well plates at a concentration of 2 × 10^5^ cells/well. They were stimulated with TNF-alpha and treated with MTX and TFB analogously to the previous tests. Supernatants were collected as follows: from the control SW-982 cell line, from the TNF-alpha-stimulated cells, and from the cell model treated with both therapeutics. They were used for subsequent chromatographic analysis.

All measurements were performed via the Dionex Ultimate 3000 LC system consisting of a quaternary pump, autosampler, and column thermostat connected to a TSQ Quantum Access Max triple quadrupole mass spectrometer (Thermo Fisher Scientific, Waltham, MA, USA). Chromatographic separation was carried out by gradient elution on an Accucore ™ RP-MS 100 × 2.1 mm, 2.6 μm particle core–shell analytical column (Thermo Fisher Scientific, MA, USA). The column was tempered at 25 °C. Mobile phases B and D consisted of 0.1% formic acid in methanol and 0.1% formic acid in methanol/water (55:45, *v*/*v*). Chromatographic separation was achieved by gradient elution: 0–6 min 20% B, 6–8 min 20–95% B, 8–15 min 95–100% B, 15–16 min 20% B, and holding at 20% B until the end. Total working time is 30 min. Thermo Xcalibur ™ (V 2.2 SP1.48) was used for system control, data collection, and processing. To detect the analyte, heated electrospray ionization (HESI) was used in the negative ionization mode with optimized parameters—prime voltage, 3000; evaporator temperature, 280 °C; enveloping gas, 40 random units; and capillary temperature, 275 °C. Deprotonated analyte molecules and internal standards were applied as precursor ions for selective reaction monitoring (SRM) with a transition of *m*/*z* 353 → 193 for 8-isoPGF2α and 357 → 197 for 8-isoPGF2α-d4. Argon was used as collision gas; the collision energy was 28 V. The concentration was calculated using the background subtraction method [30]. Additional features of the method are calibration range 25–329 pg/mL, linearity (R^2^) > 0.995, LOD = 10 pg/mL, LLOQ = 25 pg/mL, normalized matrix effect within 89.7 and 113.5%, recovery precision intra-day between 2.3 and 4.4%, inter-day between 2.4 and 5.4%, freeze–thaw stability between 89.7 and 106.6%, and post-preparative between 102.6 and 113.2%. The chromatograms with internal standard signals are presented in Appendix A.

### 4.6. Statistical Analysis

All statistical analyses were performed using GraphPad Prism (version 10, GraphPad Software, San Diego, CA, USA). Data are presented as mean ± standard deviation (SD) from at least three independent experiments. For comparisons within groups, a one-way ANOVA with a post hoc Tukey was used. A *p*-value < 0.05 was considered statistically significant. Data from the metabolic assay was processed using the Wave software 2.6.3.

## 5. Conclusions

The current study presents original results on the effect of two commonly used drugs in RA patients on a model synovial cell line. They both affect mitochondrial function in TNF-alpha-stimulated synoviocytes with MTX triggering oxidative stress and TFB partially improving mitochondrial parameters.

These results impose further research on patient-derived synoviocytes and open new perspectives for the monitoring and optimization of therapeutic protocols in RA patients.

## Figures and Tables

**Figure 1 ijms-26-08173-f001:**
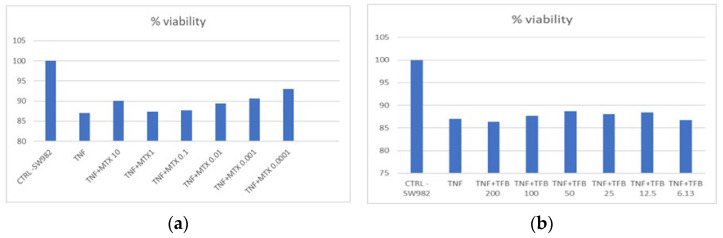
Viability (%) in TNF-alpha-stimulated SW982 cells after treatment with MTX (10, 1, 0.1, 0.1, 0.001, and 0.0001 µg/mL) (**a**) and TFB (200, 100, 50, 25, 12.5, and 6.13 Nm) (**b**).

**Figure 2 ijms-26-08173-f002:**
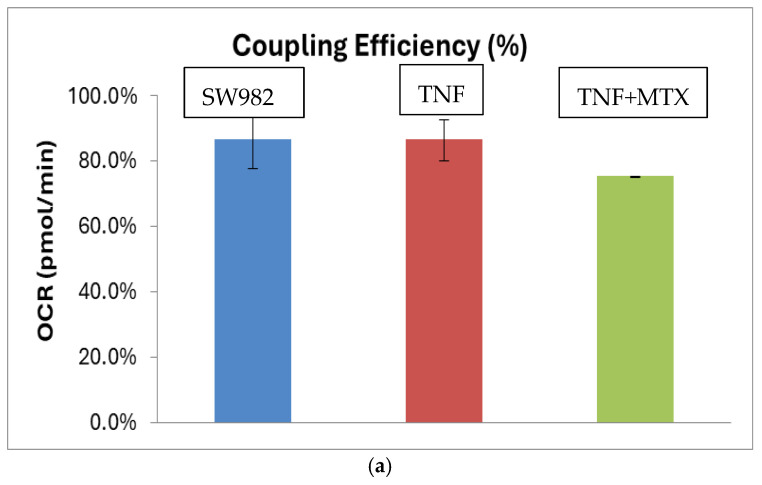
(**a**) Coupling efficiency (%) of SW982 cells. (**b**) Proton leak (pmol/min) in SW982 cells.

**Figure 3 ijms-26-08173-f003:**
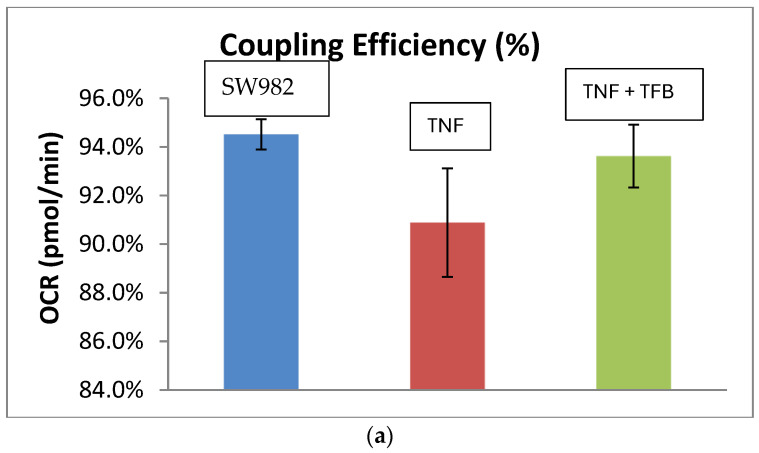
(**a**) Coupling efficiency (%) of SW982 cells. (**b**) Proton leak (pmol/min) in SW982 cells.

**Figure 4 ijms-26-08173-f004:**
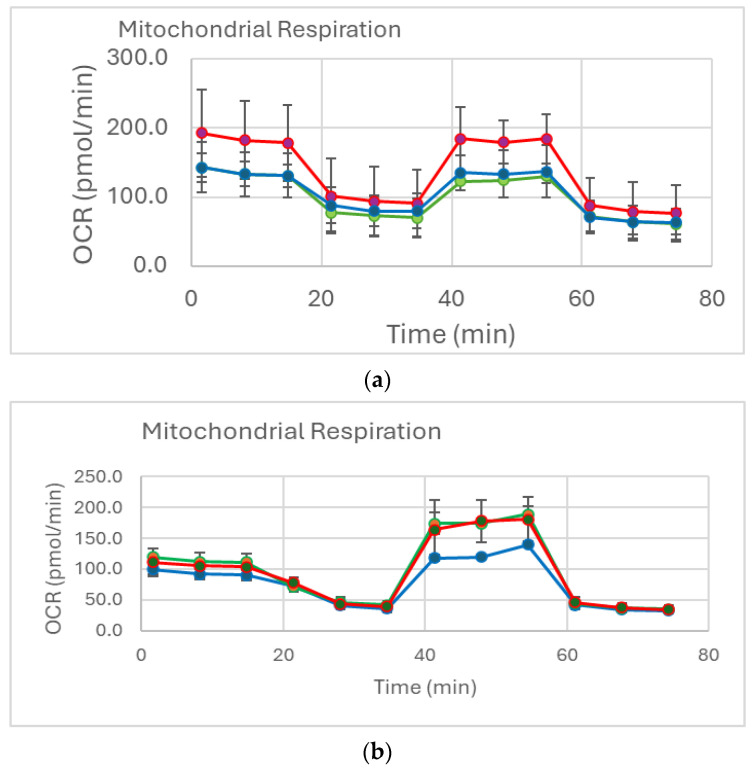
(**a**) After treated with MTX representative curves of mitochondrial respiration in SW982 cells. (**b**) After treated with TFB representative curves of mitochondrial respiration in SW982 cells.

**Figure 5 ijms-26-08173-f005:**
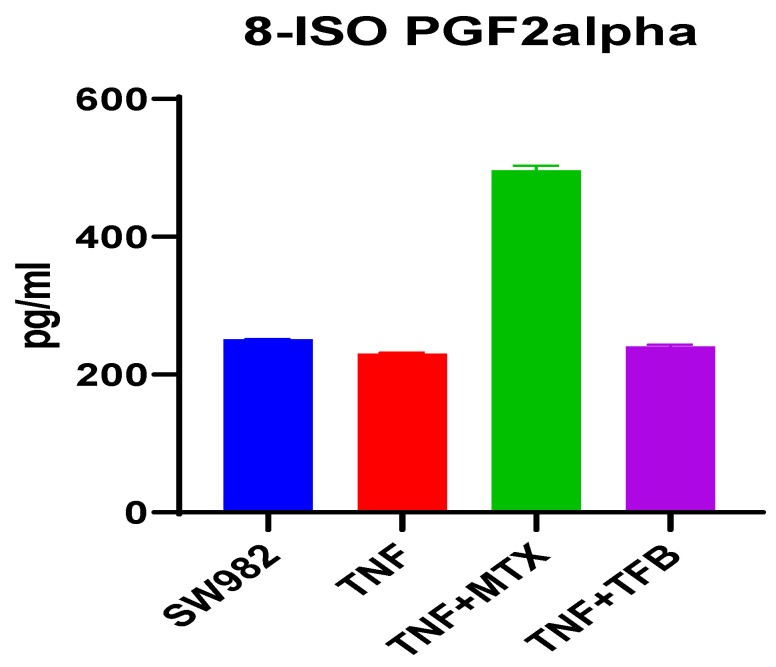
Graphic representation of 8-isoPGF2alpha levels in the supernatant of cell line SW982.

**Figure 6 ijms-26-08173-f006:**
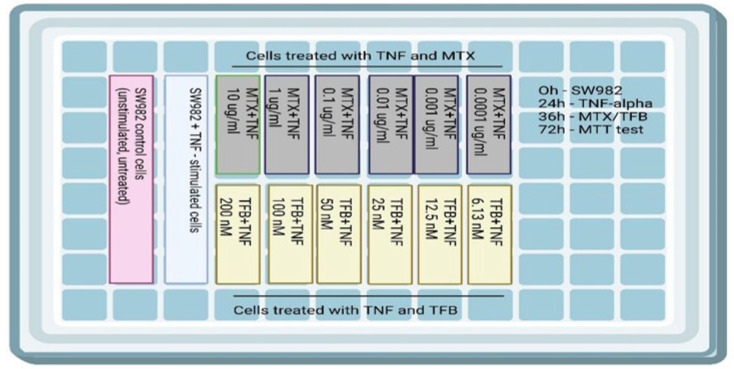
Design of the experimental flow in a 96-well culture plate.

## Data Availability

Data supporting reported results are deposited with the first author and corresponding author of the manuscript.

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
