# Peer review of "Effects of Methotrexate and Tofacitinib on Mitochondrial Function and Oxidative Stress in Human Synovial Cells In Vitro"

_ijms, 2025, doi:10.3390/ijms26178173_

Round 1
Reviewer 1 Report
Comments and Suggestions for Authors
In this manuscript, the authors investigate the effect of methotrexate and tofacitinib, two widely used RA therapeutics with distinct mechanisms of action, affect mitochondrial function and oxidative stress in an inflammatory synovial cell model (SW982). By measuring mitochondrial activity indicators such as ATP production and proton leak, along with oxidative stress levels using the marker 8-ISOPGF2A, this study provides insights into how these two medications modulate cellular energy balance and oxidative damage under TNF-alpha induced inflammation. This comparison helps to better understand how the differential impact of methotrexate and tofacitinib on mitochondrial health can inform clinical decision-making, potentially guiding personalized treatment strategies that optimize therapeutic efficacy while minimizing mitochondrial toxicity. Overall, this study exploited a well-established synovial cell model stimulating with TNF-alpha effectively mimics inflammatory conditions relevant to RA. However, several issues should be addressed.
- Introduction: please include related background information on methotrexate and tofacitinib in the context of RA, including their mechanisms of action and established roles in RA treatment. Additionally, provide a clear rationale for selecting these two drugs for comparison in the present study.
- Structure in the manuscript: Adjust the figure order so that it follows the logical flow of the manuscript. In addition, move the Materials and Methods section before the Results section to follow standard scientific reporting conventions.
- Statistical Reporting in Figures: In the results section, ensure that all figures display the corresponding p-values for statistical comparisons.
- Figure orders in the results: The current arrangement of figures in the results section is difficult to follow. Please reorganize them in a way that aligns with the narrative flow of the text, grouping related data together and ensuring that figure references in the text appear in sequential order. Moreover, the mitostress test by Seahorse is important, which should not be put in the supplementary.
- Discussion: the discussion section should provide a more comprehensive description of the study’s limitation. Now, there only two sentences. Please include other relevant information, such as the concentrations to in vivo conditions, and potential differences between the cell line used and primary synovial cells from RA patients. And RA is a whole joints disease; will these treatments affect cartilage? Moreover, please indicate the future direction of current research.
Reviewer 2 Report
Comments and Suggestions for Authors
The article "Effects of methotrexate and tofacitinib on mitochondrial function and oxidative stress in human synovial cells in vitro" aimed to evaluate methotrexate and tofacitinib on mitochondrial function and oxidative stress in TNF-alpha-stimulated SW982 synovial cells, identifying therapeutic impacts in the pathogenesis of rheumatoid arthritis. Although the paper has a certain scientific interest, several problems concerning the validity and the findings in general have been raised. The following need to be considered:
- SW982 is a synovial sarcoma line — justify relevance to RA fibroblast-like synoviocytes (FLS) and state limitations; ideally replicate in primary RA FLS.
- TNF-α 10 ng/mL ×24 h models' acute inflammation — add time-course (6, 24, 48, 72 h) and chronic exposure experiments.
- Provide PK-based justification (Cmax/synovial fluid) for 1 µg/mL MTX and 100 nM TFB; include full dose–response curves.
- Perform solvent/vehicle controls for every drug (same % DMSO or solvent) in every assay.
- MTT is dependent on mitochondrial dehydrogenases — do not use it as the only agent to select doses in mitochondria studies; complement with ATP-luminescence or trypan-blue/cell counts.
- Specify incubation conditions (L-15 medium requires ambient air — state exact COâ‚‚/temperature and humidity).
- Add STR authentication and recent mycoplasma testing for SW982.
- >2 groups comparison by multiple unpaired t-tests is invalid — instead, use one-way (or two-way) ANOVA with appropriate post-hoc (Tukey/Dunnett) and correct for multiple comparisons.
- Report exact p-values and 95% CIs; graph stars consistently and avoid ad-hoc cut-offs (e.g., p=0.0018 highlighted **).
- Test for normality; if broken, use non-parametric tests (Kruskal–Wallis with Dunn post-hoc).
- Report effect sizes (mean differences) and confidence intervals — do not rely on p-value alone.
- Instrument model, seeding density per well (cells/well), final oligomycin/FCCP/rotenone+antimycin A concentrations, and injection order, normalization (protein/DNA/cell number).
- Include calibration range, LOD/LOQ, linearity (R²), recovery, matrix effects, intra/inter-day precision, stability, and chromatograms with internal standard signal.
Round 2
Reviewer 1 Report
Comments and Suggestions for Authors
Thanks for revising the manuscript. I do not have any further questions.
Reviewer 2 Report
Comments and Suggestions for Authors
After thoroughly reviewing the revised manuscript and considering the authors' revisions and responses to the referee's comments, I find that the manuscript has been significantly improved. The authors have effectively addressed the concerns, thereby enhancing the clarity and scientific rigour of their study. The revisions have clarified the methodology, improved the presentation of results, and strengthened the discussion and conclusions.
Therefore, I believe that the manuscript now meets the standards required for publication in IJMS, and I recommend that it be accepted for publication.